# An Investigation of Potential Sources of Nutraceuticals from the Niger Delta Areas, Nigeria for Attenuating Oxidative Stress

**DOI:** 10.3390/medicines6010015

**Published:** 2019-01-20

**Authors:** Lucky Legbosi Nwidu, Philip Cheriose Nzien Alikwe, Ekramy Elmorsy, Wayne Grant Carter

**Affiliations:** 1Department of Experimental Pharmacology and Toxicology, Faculty of Pharmaceutical Sciences, University of Port Harcourt, Port Harcourt PMB 5323, Rivers State, Nigeria; 2School of Medicine, University of Nottingham, Royal Derby Hospital Centre, Derby DE22 3DT, UK; ekramy_elmorsy@yahoo.com (E.E.); wayne.carter@nottingham.ac.uk (W.G.C.); 3Department of Animal Science, Niger Delta University, Wilberforce Island, Yenegoa PMB 071, Bayelsa State, Nigeria; bushdoctor2013@gmail.com; 4Department of Forensic Medicine and Clinical Toxicology, Faculty of Medicine, Mansoura University, Mansoura 35516, Egypt

**Keywords:** acetylcholinesterase inhibitor, antioxidants, memory enhancers, nutraceuticals

## Abstract

**Background:** Diets rich in fruits, vegetables, and medicinal plants possess antioxidants potentially capable of mitigating cellular oxidative stress. This study investigated the antioxidant, anti-acetylcholinesterase (AChE), and total phenolic and flavonoids contents (TPC/TFC) of dietary sources traditionally used for memory enhancing in Niger Delta, Nigeria. **Methods:**
*Dacroydes edulis *methanolic seed extract (DEMSE), *Cola lepidota* methanolic seed extract (CLMSE), *Terminalia catappa* methanolic seed extract (TeCMSE), *Tricosanthes cucumerina* methanolic seed extract (TrCMSE), *Tetrapleura tetraptera* methanolic seed extract (TTMSE), and defatted *Moringa oleifera* methanolic seed extract (DMOMSE); *Dennettia tripetala* methanolic fruit extract (DTMFE), *Artocarpus communis* methanolic fruit extract (ACMFE), *Gnetum africana* methanolic leaf extract (GAMLE), *Musa paradisiaca* methanolic stembark extract (MPMSE), and *Mangifera indica* methanolic stembark extract (MIMSE) were evaluated for free radical scavenging antioxidant ability using 2,2-Diphenyl-1-picrylhydrazyl (DPPH), reducing power capacity (reduction of ferric iron to ferrous iron), AChE inhibitory potential by Ellman assay, and then TPC/TFC contents determined by estimating milli-equivalents of Gallic acid and Quercetin per gram, respectively. **Results:** The radical scavenging percentages were as follows: MIMSE (58%), MPMSE (50%), TrCMSE (42%), GAMLE (40%), CLMSE (40%), DMOMSE (38%), and DEMFE (37%) relative to β-tocopherol (98%). The highest iron reducing (antioxidant) capacity was by TrCMSE (52%), MIMSE (40%) and GAMLE (38%). Extracts of MIMSE, TrCMSE, DTMFE, TTMSE, and CLMSE exhibited concentration-dependent AChE inhibitory activity (*p* < 0.05–0.001). At a concentration of 200 µg/mL, the AChE inhibitory activity and IC_50_ (µg/mL) exhibited by the most potent extracts were: MIMSE (≈50%/111.9), TrCMSE (≈47%/201.2), DTMFE (≈32%/529.9), TTMSE (≈26%/495.4), and CLMSE (≈25%/438.4). The highest TPC were from MIMSE (156.2), TrCMSE (132.65), GAMLE (123.26), and CLMSE (119.63) in mg gallic acid equivalents/g, and for TFC were: MISME (87.35), GAMLE (73.26), ACMFE (69.54), CLMSE (68.35), and TCMSE2 (64.34) mg quercetin equivalents/gram. **Conclusions**: The results suggest that certain inedible and edible foodstuffs, most notably MIMSE, MPMSE, TrCMSE, GAMLE, and CLMSE may be beneficial to ameliorate the potentially damaging effects of redox stress.

## 1. Introduction

Oxidative stress is associated with a number of diseases and arises as a consequence of an imbalance between reactive oxygen species (ROS), reactive nitrogen species (RNS), and their dissipation via enzymatic and non-enzymatic mechanisms [1]. Many factors including UV-irradiation, industrial emissions, tobacco smoke, licit or illicit drug usage, heavy metal exposure, inorganic and organic contaminants, and xenobiotics are among the potential exogenous sources of ROS generation in regions such as the Niger Delta, Nigeria. Endogenous sources of ROS include those generated via normal cellular metabolism and via pathological means [2]. The chronic exposure to exogenous factors and/or activation of endogenous means can provoke oxidative insults that may activate stress response pathways including inflammation, cytokine secretion, and apoptosis that might contribute to wide range of pathophysiological events [1,3]. 

Nutraceuticals are used as dietary supplements, but also as auxiliaries for the perceived prevention and/or treatment of a variety of diseases and disorders. The role of consumption of nutraceuticals and their protective effects in mammals and humans against diseases such as neurodegeneration that includes an elevation of oxidative stress have been reviewed [4,5]. Natural antioxidants in foods may exhibit protective antioxidant effects and thereby aid in the reduction of premature mortality [6,7]. One such group of antioxidants is flavonoids, compounds that are ubiquitous in many edible plants [8]. Collectively, there are an extensive number of low molecular weight organic compounds, including polyphenols and flavonoids that have been termed secondary metabolites or phytochemicals that can be specific to each plant. These arrays of secondary metabolites can exhibit a wide spectrum of pharmacological effects including provision of cellular antioxidant activities capable of scavenging damaging free radicals. This endowed useful activity provides potential functional benefits to humans beyond basic nutrition, and could be exploited as commercial sources of nutraceutical formulations [9]. These potential health benefits provide an impetus for subjecting plant extracts and fractions for scrutiny to elucidate and quantify their respective antioxidant and other health benefiting abilities.

In the Niger Delta region of Nigeria residents may benefit from the wide patronage and chronic consumption of numerous endogenous edible seeds, fruits, nuts, pods, green leafy vegetables, herbs, spices, and crops. These are commonly consumed in either raw or in cooked forms in various cuisines. Many of these foodstuffs have yet to be evaluated for their ability to mitigate oxidative stress. Additionally, there is an association of oxidative stress and a cholinergic deficit with neurodegenerative diseases such as Alzheimer’s disease and Parkinson’s diseases [10,11,12,13,14]. Hence, the intake of appropriate foodstuffs able to combat this cellular damage and loss of neuronal functionality may limit the development or indeed propagation of neurodegenerative disease [15,16,17,18]. 

Therefore, this study investigated extracts of *Dacroydes edulis* methanolic seed extract (DEMSE), *Cola lepidota* methanolic seed extract (CLMSE), *Terminalia catappa* methanolic seed extract (TeCMSE), *Tricosanthes cucumerina* methanolic seed extract (TrCMSE), *Tetrapleura tetraptera* methanolic seed extract (TTMSE), defatted *Moringa oleifera* methanolic seed extract (DMOMSE); *Dennettia tripetala* methanolic fruit extract (DTMFE), *Artocarpus communis* methanolic fruit extract (ACMFE), *Gnetum africana* methanolic leaf extract (GAMLE), *Musa paradisiaca* methanolic stembark extract (MPMSE), and *Mangifera indica* methanolic stembark extract (MIMSE) for in vitro antioxidant and anti-acetylcholinesterase effects and associated polyphenolic and flavonoid contents. 

*Dacryodes edulis* G. Don Lam (Burseraceae) is an edible pear native to the tropics. In the Niger Delta, the fruit is boiled or softened by exposure to heat and used to eat Zea mays (maize) or guinea corn. The pulp may also be boiled or roasted to form a kind of butter [19,20]. *D. edulis* leaf, fruit, and resin extracts have numerous pharmacological activities including antioxidant [21], anti-microbial [22], and anti-carcinogenic [23] properties. 

*Cola lepidota* (Sterculiaceae) is popularly known as monkey cola. The plant is indigenous to tropical Africa and has its center of greatest diversity in West Africa [24]. The native peoples of southern Nigeria and the Cameron relish the fruits as a source of foodstuffs. Seeds of the monkey cola species are not edible, unlike the seeds of kola nut (*C. nitida*). *C. lepidota* is used in traditional medicine with functions that include its use as a stimulant, and to suppress sleep and for pulmonary problems and cancer-related ailments [25,26], with seed and fruit pulp extracts also displaying antioxidant activity [27]. Phytochemical analysis of the plant included detection of flavonoids [28]. 

*Trichosanthes cucumerina* Linn (Cucurbitaceae) is an annual, dioecious climber, widely distributed in Asian countries [29]. *T. cucumerina* fruit is consumed as a vegetable by rural dwellers, especially in the Western part of Africa. It is commonly called snake gourd, viper gourd, snake tomato, or long tomato [30]. The fruit is used as a cathartic, the seeds used for stomach disorders, anti-febrile and anti-helmintic activities and cardioprotective activities have also been reported [31].

*Dennettia tripetala* G. Baker (Annonaceae) is a fruit used as a spice and condiment in West Africa [32]. *D. tripetala* fruit is used in ethnomedicine to treat cold, fever, typhoid, cough, worm infestation, vomiting, stomach upset, and as an appetite enhancer [33]. Strong anti-nociceptive effects comparable to opioid agonists and non-steroidal anti-inflammatory drugs have been demonstrated [33]. Tannins, terpenoids, and other phytochemicals of *D. tripetala* are reported to be responsible for wide range of bioactivities [34].

*Artocarpus comminis* (Moraceae) is a flowering tree from the mulberry family. It is locally called breadfruit tree because of the "bread-like texture" of its edible fruits. *A. comminis* grown in the Niger Delta region is extensively used as both a food and traditional medicine. Studies have shown that *A. communis* possesses several bioactivities, such as antioxidant [35], anti-cancer [36,37], and anti-inflammatory activities [38,39]. Biologically active phytochemicals within *A. communis* include flavonoids, chalcones, and stilbenes [40]. 

*Terminalia catappa* Linn. (Combretaceae) is native to Southeast Asia. It is widely planted throughout the tropics and the Niger Delta region of Nigeria. *T. catappa* nut kernel can be eaten raw [41]. The ethnopharmacological properties of this plant have yet to be fully evaluated.

*Moringa oleifera* (Moringaceae), commonly known as horseradish tree or drumstick tree, is widely cultivated in Africa and other regions including South East Asia, and is considered a multi-purpose plant [42]. There are a broad number of bioactive agents present within the seeds, and for which seed extracts have been reported to exhibit neuroprotective effects [43,44,45]. 

*Tetrapleura tetraptera* (Mimosaceae), locally referred to as “Arindan” in Yoruba, is a flowering plant of the pea family native to West Africa. The dried fruit is used as a seasoning spice in the Southern part of Nigeria [46,47] and in the Niger Delta areas, the pod, fruit, and seeds are used as spices. A neuroprotective effect of *T. tetraptera* has been reported in scopolamine-induced amnesic rats [48].

*Gnetum africanum* (Gnetaceae) leaf, also known as wild spinach, is utilized as a food [49]. The vegetable is locally known under different nomenclature: sorgo (in Ogoni in Rivers state, Nigeria), afang (Ibibios in South-South, Nigeria), okazi (Igbo in South-East Nigeria), okok or eru (Cameroun), and fumbwa (Democratic Republic of Congo). Leaves are eaten as a vegetable raw or cooked and revered for their nutritional and therapeutic properties. The vegetable is domesticated for its economic potential, and useful component of dietary fibre, essential amino acids, vitamins, and minerals [50].

*Musa paradisiaca* (Musaceae), a banana plant, has a number of reported medicinal uses, for example banana seed mucilage has been applied as a treatment for catarrh and diarrhea [51]. There is also an extensive list of reported pharmacological activities including hepatoprotective effects [51].

*Mangifera indica* (Anacardiaceae) aqueous stem bark extract has been utilized as a remedy for diarrhea, fever, gastritis, and ulcers [52]. It has a number of biological activities, including the ability to act as an anti-cancer, and anti-bacterial agent [53]. 

Collectively, the plants described above were among the commonly edible and non-edible components of daily diets of Niger Deltans, hence an investigation of their relative antioxidant and anti-cholinesterase activities were undertaken.

## 2. Materials and Methods 

### 2.1. Collection and Identification of Plant Materials

The seeds of *Dacroydes edulis, Cola lepidota, Tricosanthes cucumerina, Terminalia catappa, Tetrapleura tetraptera*, and defatted *Moringa oleifera* seed; fruits of *Dennettia tripetala, Artocarpus communis*, green leafy vegetable of *Gnetum africana*; stembark of *Musa paradisiaca* and *Mangifera indica* were collected from Niger Delta University Agricultural Extension farm, Amassoma, Yenegoa, Bayelsa State, Nigeria in March 2015 and authenticated by Mr. Philip Cheriose Nzien Alikwe, an Agriculturalist from the Department of Animal Science, Niger Delta University, Wilberforce Island, Bayelsa State, Nigeria. The voucher numbers (NDUH/P/71-81) were deposited in the University’s herbarium, Niger Delta University, Nigeria. All of the samples were shade or air-dried for seven days and powdered using an electrical blender.

### 2.2. Preparation of Hydromethanolic Extracts 

The seeds of *Dacroydes edulis*, *Cola lepidota, Tricosanthes cucumerina, Terminalia catappa, Tetrapleura tetraptera*, and defatted *Moringa oleifera* seed; fruits of *Dennettia tripetala*, *Artocarpus communis* and leafy vegetable of *Gnetum africanum*; stembark of *Musa paradisiaca* and *Mangifera indica* were powdered, and 10 g of each were macerated in 100 mL of 50% methanol for 72 h with vigorous hand agitation for one minute three times daily. Double layered gauze was used for filtration to obtain filtrates that were then reduced in volume at 40 °C on a water bath (Model TT6, Techmel and Techmel, Asaba, Nigeria) to obtain dried extracts. The extracts were weighed and the yield recovered (as a percentage) and recorded (Table 1). Extracts were stored in a refrigerator at 4 °C in an airtight container until used for experimental studies.

### 2.3. Chemicals

Acetylthiocholine iodide (ATCI), L-ascorbic acid, bovine serum albumin (BSA), 2, 2-Diphenyl-1-picrylhydrazyl (DPPH), 5,5-dithiobis [2-nitrobenzoic acid] (DTNB), Folin-Ciocalteu reagent, physostigmine, and β-tocopherol were all purchased from Sigma Aldrich (Poole, UK), as were all the other chemicals used unless stated otherwise.

### 2.4. Animals 

Rat brain homogenates from male F344 strain rats (200–230 g) were utilized as source of mammalian AChE, as described in an earlier report [54]. Rats were maintained at a controlled temperature of 21 ± 1 °C and a cycle of 16 h light/8 h dark with food intake daily and water ad libitum. Approval for the use of animals was obtained from the University of Nottingham Local Ethical Review Committee (study reference CHE 10, project licence approval code: PPL: 40/2624, approval date 13 June 2005 and the study was executed in line with the Animals Scientific Procedures Act (UK) 1986.

### 2.5. 2,2-Diphenyl-1-Picrylhydrazyl (DPPH) Radical Scavenging Effect

Spectrophotometric assays that utilized DPPH radical scavenging were used to quantify antioxidant activity. DPPH has been utilized extensively as a stable organic radical to evaluate scavenging activities of plethora of natural compounds such as flavonoids, polyphenols, and crude plant extracts and fractions. Antioxidants scavenge DPPH radicals by donating an electron to form reduced DPPH changing the colour of the solution from purple to yellow, the level of which can be quantified by spectrophotometry. DPPH radical scavenging assays were performed according to the method of Nwidu et al. [54]. Stock solutions of the plant extracts (5 mg/mL) were diluted to final concentrations of 200, 100, 50, 25, 12.5, and 6.25 µg/mL in ethanol. Then, 160 µL of 0.1 mM DPPH in ethanol solution was added to 20 µL solutions of the extracts as well as a standard and 20 µL H_2_O. For the standard, *β*-tocopherol was prepared at concentrations of 1.56, 0.78, 0.39, 0.195, and 0.0975 mg/mL. Assays were performed at 37 °C for 40 min in the dark, and thereafter the absorbance was read at 517 nm, as described in a previous report [54]. All reactions were performed in triplicates, from which an average was generated.

### 2.6. Reducing Power Capacity Assay

The reducing capacity (antioxidant ability) of the plant extract was also estimated based on its ability to reduce ferric ions (Fe^3+)^ to ferrous ions (Fe^2+^). The plant extracts were assayed over the concentration ranged of 6.25–50 µg/mL. Four µL of 5 mg/mL of each plant extract was mixed with 400 µL of phosphate buffer (0.2 M dibasic sodium phosphate and 0.2 M monobasic sodium phosphate buffer adjusted to pH 7.4), 250 µL of 1% potassium ferricyanide was then added and the mixture incubated at 50 °C for 20 min. Then, 250 µL of 10% trichloroacetic acid was added, and after mixing, the solution was centrifuged at 3000 rpm for 10 min. One hundred µL from the supernatant was mixed with an equal volume of water, followed by 20 µL of freshly prepared ferric chloride solution. After mixing, the absorbance was measured at 700 nm in a microtiter plate reader, as previously reported [54]. Ascorbic acid was the reference substrate and the following concentration range was employed (0.3, 0.6, 0.9, 1.2, 1.5, and 3 mg/mL). All reactions were performed in triplicates, from which an average reading was generated.

### 2.7. Acetylcholinesterase Inhibition Assay

The assay for AChE inhibition was based upon the method of Ellman et al. [55], but modified for a 96-well microtiter plate format, as reported in Nwidu et al. [54]. In a microtiter plate, 40 μL of plant extract (at concentrations of 200, 20, 2, 0.2 and 0.02 µg/mL) was mixed with 35 µL of 50 mM Tris-HCl (pH 8.0) containing 0.1% BSA, 50 μL of 3 mM DTNB, and 50 µL of AChE. The AChE used was either from electric eel at 1 mg/mL (Sigma, Poole, UK) or that present within rat brain homogenate (prepared at 10% (*w*/*v*), according to the procedure of Carter et al. [56,57], which had been diluted 1:10 in 10 mM Tris-HCl pH 8.0 for assays. Plates were incubated at 37 °C for 5 min before the cholinesterase reaction initiated by the addition of 25 μL of 15 mM ATCI substrate, resulting in the production of 5-thio-2-nitrobenzoate anion that was read at 412 nm every 5 s for 10 min using a Spectramax microplate reader (Thermo Fisher, Stafford, UK). Eserine was employed at 0.02 µg/mL as a positive control for AChE inhibition. At this concentration (or above), eserine inhibits AChE to ≈100% [55]. All reactions were performed in triplicates, from which an average reading was generated.

### 2.8. Determination of Total Phenolic Content

A Folin-Ciocalteu Reagent (FCR) spectrophotometric method was used to quantify the total phenolic content in plant extracts, as described previously [54]. Twenty µL of each concentration of the plant extracts (ranging from 1–100 µg/mL) was added to 90 µL of water followed by addition of 30 µL of FCR, and samples were vigorously shaken within a microtiter plate reader. Within 30 s and a total assay time of eight minutes, 60 µL of 7.5% Na_2_CO_3_ solution was added to each microtiter well and then plates were incubated at 40 °C on a shaking incubator. The absorbance of the mixture was read after 40 min at 760 nm, as detailed in a previous report [56]. Gallic acid was used as the positive control substance. All reactions were performed in triplicates, from which an average reading was generated.

### 2.9. Determination of Total Flavonoid Content

Total flavonoid contents of the plant extracts were determined according to the method described by Nwidu et al. [54] using quercetin as a reference compound. Twenty microliter of plant extracts (5 mg/mL) were dissolved in ethanol and then mixed with 200 μL of 10% aluminum chloride solution and 1 M potassium acetate solution in microtiter plate wells. Samples were incubated for 30 min at room temperature, after which the absorbance of the solution was measured at 415 nm, as reported earlier [54]. Quercetin was used as the reference compound. All reactions were performed in triplicates, from which an average reading was generated.

### 2.10. Statistical Analysis

Results are expressed as the mean ± SD. IC_50_ values for each extract or fraction were calculated using non-linear regression analysis. A Spearman rank-order correlation coefficient was used to assess the relationship between total phenolic content, total flavonoid content, antioxidant content, and inhibition of AChE activity. Statistical analyses were performed using GraphPad Prism (Version 5.3) for Windows (GraphPad Software, Inc., San Diego, CA, USA, www.graphpad.com). A *p* value of <0.05 for results was considered to be statistically significant.

## 3. Results

### 3.1. DPPH Radical Scavenging Activity

Aqueous methanolic extracts of the inedible stem-bark, edible fruits, seeds and leaf extracts displayed DPPH radical scavenging activities in a concentrations-dependent manner as shown in Figure 1. From these analyses IC_50_ values for radical scavenging were calculated with results displayed in Table 1. When assessed at a concentration of 1000 µg/mL, the majority of aqueous methanolic extracts displayed significant (*p* < 0.05–0.001) DPPH radical scavenging effects, that ranged from ≈38–58% of that observed with Vitamin E (set at 100%). For the tested extracts, MIMSE (IC_50_ = 321 µg/mL) and MPMSE (IC_50_ = 106 µg/mL) demonstrated the highest percent inhibitions of 58% and 50%, respectively, and the latter extract was also the most potent (lowest IC_50_ value). Collectively, the descending order of DPPH radical scavenging activity was: MIMSE > MPMSE > TrCMSE > GAMLE > CLMSE > DMOMSE > DEMSE > DTMSE > ACMFE > TeCMSE > TTMSE with radical scavenging percentages of 58%, 50%, 42%, 40%, 40%, 38%, 37%, 21%, 20%, 19%, and 18%, respectively. The descending orders of potency of the extracts as radical scavengers as determined via IC_50_ values (µg/mL) were: MPMSE > DTMFE > DEMSE > DMOMSE > TTMSE > TeCMSE > MIMSE > CLMSE > GAMLE > TrCMSE > ACMFE (Table 1).

### 3.2. Reducing (Antioxidant) Capacity 

An evaluation of the reducing capacity of the aqueous methanolic extracts from the edible and non-edible foods showed that these also displayed antioxidant abilities in a concentration-dependent manner (Figure 2).

At a concentration of 50 µg/mL, all of the extracts demonstrated significant antioxidant effects except for TeCMSE and DTMFE when compared with Vitamin C (ascorbic acid). The highest antioxidant capacity was demonstrated by TCMSE2 (52%), MIMSE (40%) and then GAMLE (38%) relative to Vitamin C at 100%. The order of descending reducing capacity for the extracts was: TrCMSE > MIMSE > GAMLE > DEMSE > DMOMSE > ACMFE > CLMSE > MPMSE > DTMFE > TeCMSE.

### 3.3. Acetylcholinesterase Inhibitory Activity 

Methanolic aqueous extracts of the inedible stem-bark, edible fruits, seeds, and leaf extracts of the evaluated plants displayed concentrations-dependent AChE inhibition, as shown in Figure 3. 

Across the investigated concentrations the level of AChE inhibition was used to generate IC_50_ concentrations (Table 1). 

At the higher concentrations assayed, all evaluated extracts exhibited significant (*p* < 0.001) concentration dependent AChE inhibitory activity, with percentage AChE inhibitions at 200 µg/mL ranging from ≈17–50%. The descending order of AChE inhibitory activity for the extracts was MISME > TrCMSE > DEMSE > PPMS >TTMS > CLMSE > DMOMSE > MPMSE > GAMLE > ACMFE > TeCMSE. The descending order of potency, as determined by IC_50_ values, were: MIMSE > TrCMSE > GAMLE > CLMSE > DEMSE > ACMFE > MPMSE > DTMSE > DMOMSE > TeCMSE > TTMSE.

### 3.4. Total Phenolic Content and Total Flavonoid Content

Total phenolic and total flavonoid contents were determined for each of the extracts and these have been included in Table 1. The relatively higher TPC levels (above 100 mg GAE/g) were observed for MISME, TrCMSE, GAMLE, CLMSE, and ACMFE at 156.2, 132.65, 123.26, 119.63, and 102.45 mg GAE/g, respectively. The relatively higher TFC levels (above 50 mg QUER/g) were recorded with MIMSE, GAMLE, ACMFE, CLMSE, TrCMSE, and DEMSE at 87.35, 73.26, 69.54, 68.35, 64.34, and 53.35 mg QUER/g, respectively.

### 3.5. Correlation between AChE Inhibition, Antioxidant Ability, and Total Phenolic and Flavonoid Contents

To consider if there was a relationship between AChE inhibition potency or DPPH radical scavenging potency and total phenolic or flavonoid content, Spearman rank correlations were calculated. The correlation coefficients (*R*-values) and significance of association (*p*-values) are shown in Table 2. The ability of extracts to inhibit AChE (measured as increasing IC_50_ values i.e., reduced potency) was significantly inversely correlated with increasing phenolic or increasing flavonoid content. Hence, extracts that displayed relatively high AChE inhibitory activity also retained relatively high phenolic or flavonoid content. By comparison, there was a positive but non-significant correlation between AChE inhibitory potency and either DPPH radical potency, or total phenolic or flavonoid content (Table 2).

## 4. Discussion

Medicinal plants, spices, fruits, seeds, or vegetables provide an array of chemical entities with therapeutic potential. For example, medicinal plants may provide antioxidants in the form of flavonoids or polyphenols that are valuable assets for protection against oxidative stress and associated diseases. The public and scientific interest regarding the utilization of natural antioxidants continues to grow due to their potential or indeed perceived health-promoting effects. 

Our analyses have shown *M. parasidisiaca* (106 µg/mL), *D. tripetala* (136 µg/mL), defatted *M. oleifera* (138 µg/mL), *T. tetraptera* (205 µg/mL), *T. catappa* (302 µg/mL) and *M. indica* (321 µg/mL) have highly active and significant DPPH (IC_50_) radical scavenging abilities. Other independent studies have also reported antioxidant properties of *M. parasidisiaca* [58,59,60], *D. tripetala* [61] *M. oleifera* [62,63], *T. tetraptera* [64,65], *T. catappa* [66], *M. indica* [67,68], and *T. cucumenina* [69,70]. Additionally, the protective effect of a natural extract from the stem-bark of *M. indica* was able to counter age-associated oxidative stress in elderly humans, indicative of its potential to act as a nutraceutical in vivo [68]. The antioxidant effects of the fruit of *Artrocarpus communis* [71] and the leaves of *Dacroydes edulis* [72] have also been reported. Interestingly, fruit (*Dacroydes edulis)* and vegetable (*Gnetum africanum*) intake and an imbalance of oxidant/antioxidant status was reported to be associated with the development of diabetic retinopathy [72], with a recommendation that a diet rich in antioxidant supplements and tight glycemic control could postpone the onset of diabetic retinopathy [72]. 

The assessment of diets in a number of epidemiological studies and via quantitative evaluation have suggested that adherence to a Mediterranean-style diet and diets rich in fruits and vegetables may have protective benefits against age-related cognitive decline and neurodegenerative diseases [73,74,75,76,77,78,79,80,81,82]. This led us to consider the acetylcholinesterase inhibitory activity of these plants, since cholinesterase inhibitors are the mainstay of treatment for mild to moderate AD. An assessment of the potencies of a broad number of plant anticholinesterases inhibitors has been undertaken [83,84], with IC_50_ values ranging from 0.3 to 100.4 µg/mL. Hence, the plant extracts analyzed herein only displayed mild or moderate anti-AChE activities, with *M. indica* the most potent (IC_50_ of 111.9 µg/mL). Nevertheless, although only relatively weak cholinesterase inhibitors per se, chronic consumption of these foodstuffs might still provide provision of chemical entities able to ameliorate development or propagation of neurodegenerative disease. Indeed, an aqueous decoction of mango (*Mangifera indica* L.) stem bark has been developed on an industrial scale to be used as a nutritional supplement, cosmetic, and as a nutraceutical with neuroprotective effects [68,85,86]. Furthermore, neuroprotective effects of *Moringa oleifera* seed extract [52,53] and likewise *T. tetraptera* have also been demonstrated [48]. 

Our study also quantified the levels of phenolics and flavonoids, as these phytochemicals are widely distributed in the plant kingdom and possess antioxidant and anti-inflammatory activities [87]. Many of the active extracts investigated possessed high polyphenols and flavonoids content (Table 1) comparable to gallic acid and quercetin, respectively. Certain dietary phytochemicals, such as polyphenols have been reported to possess potential protection of cognitive function during aging [88] and may serve as natural neuroprotective agents [89,90,91]. In addition to their action as neuroprotective agents, flavonoids may also be efficacious candidates as potential pharmaceuticals or nutraceuticals for the treatment of AD [92]. Antioxidant activities of green tea phytochemicals and nutraceuticals such as curcumin, catechins, licopene, resveratrol, piperine, and anthocyanins, have been reported using in vitro and in vivo models [93,94].

Of interest, there was a significant inverse correlation between the potency of AChE inhibition (IC_50_ values) and total phenolic or flavonoid contents. This suggests that the agent(s) responsible for the AChE inhibitory activity are resident within the phenolic and flavonoid compounds. By contrast, there was no significant correlation between the AChE inhibition potency and that for DPPH radical scavenging, suggesting that the agent(s) that provide AChE inhibition is different from that for radical scavenging. Likewise, there was no correlation between AChE inhibitory potency and antioxidant activity, or between antioxidant activity and TPC or TFC, hence the chemical agent(s) that provided antioxidant protection were not AChE inhibitors, or likely to be abundant polyphenols or flavonoids. 

A clear limitation of our study is that we have only assessed in vitro properties of these plant parts and their respective polyphenol and flavonoid content. We are unable to directly comment on how much of these foodstuffs are typically eaten, and indeed this will vary extensively between peoples and their food preparation methods. However, irrespective of these limitations, it is provocative to propose that a suitable diet rich in certain phytochemicals may provide beneficial counter-measures against oxidative stress-induced damage and its impact upon disease pathogenesis and propagation.

## Figures and Tables

**Figure 1 medicines-06-00015-f001:**
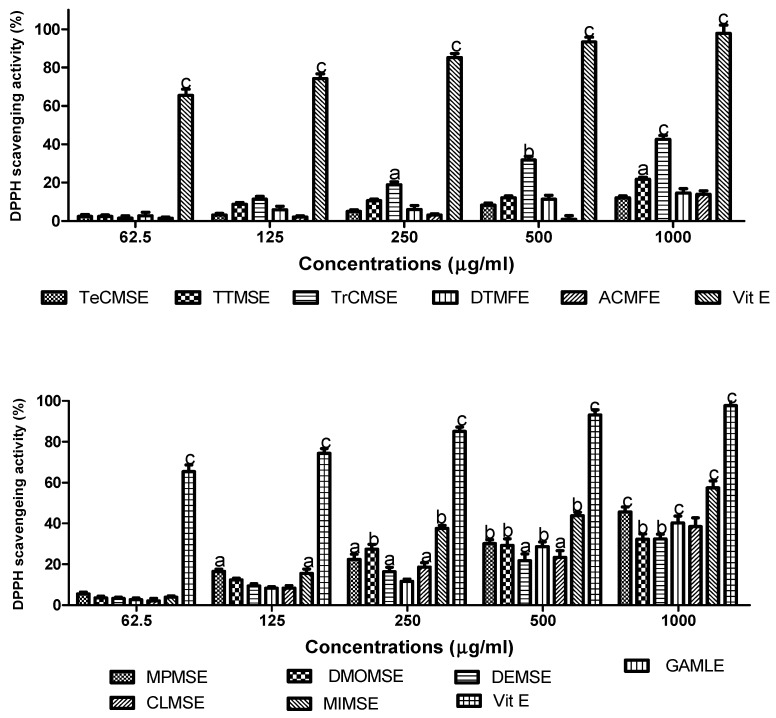
DPPH radical scavenging activity of plant extracts. Plant antioxidant activity was measured via percentage inhibition of radical scavenging of DPPH. Results are expressed as means ± SEM for three separate experiments at each concentration. TeCMSE, *Terminalia catappa* methanolic seed extract; TTMSE, *Tetrapleura tetraptera* methanolic seed extract; TrCMSE, *Tricosanthes cucumerina* methanolic seed extract; DTMFE, *Dennettia tripetala* methanolic fruit extract; ACMFE, *Artocarpus communis* methanolic fruit extract; MPMSE, *Musa parasidisiaca* methanolic stem-bark extract; DMOMSE, Defatted *Moringa oleifera* methanolic seed extract; DEMSE, *Dacroydes edulis* methanolic seed extract; GAMLE, *Gnetum africanum* methanolic leaf extract; CLMSE, *Cola lepidota* methanolic seed extract; MIMSE, *Mangifera indica* methanolic stem-bark extract; Vit E, Vitamin E (β-Tocopherol). Results are expressed as means ± SEM for three separate experiments at each concentration. For marked significance from controls, a: *p* < 0.05, b: *p* < 0.01, c: *p* < 0.001.

**Figure 2 medicines-06-00015-f002:**
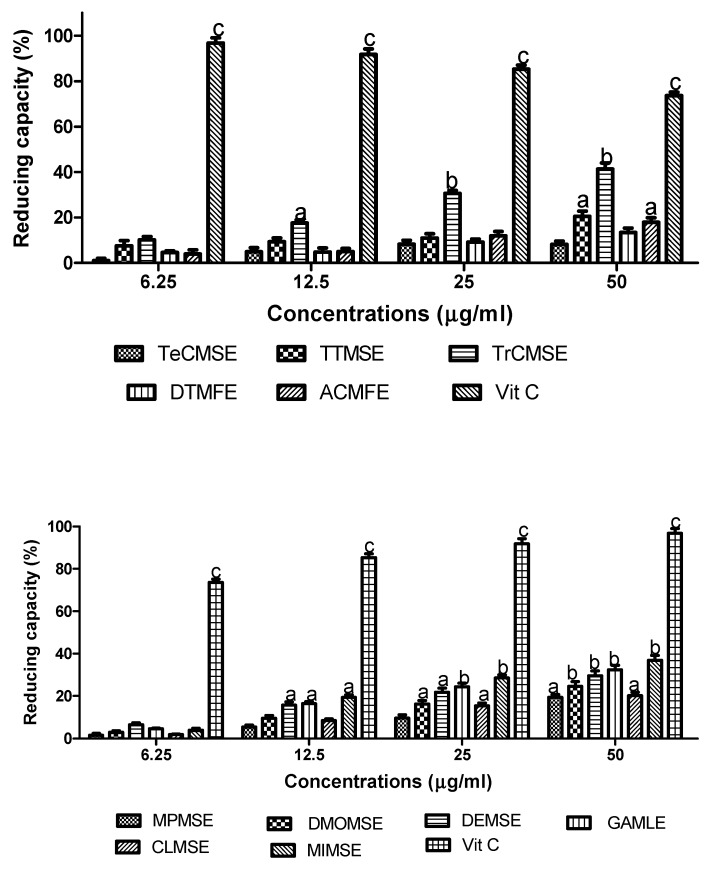
Reducing capacity of plant extracts. Plant reducing power was assessed via the ability to reduce ferric (Fe^3+^) to ferrous (Fe^2+^) iron. The percentage increase of reductive capacity with increasing plant extract concentration was determined. Vitamin C was used as a positive control. Results are expressed as means ± SEM for three separate experiments at each concentration. TeCMSE, *Terminalia catappa* methanolic seed extract; TTMSE, *Tetrapleura tetraptera* methanolic seed extract; TrCMSE, *Tricosanthes cucumerina* methanolic seed extract; DTMFE, *Dennettia tripetala* methanolic fruit extract; ACMFE, *Artocarpus communis* methanolic fruit extract; MPMSE, *Musa parasidisiaca* methanolic stem-bark extract; DMOMSE, Defatted *Moringa oleifera* methanolic seed extract; DEMSE, *Dacroydes edulis* methanolic seed extract; GAMLE, *Gnetum africanum* methanolic leaf extract; CLMSE, *Cola lepidota* methanolic seed extract; MIMSE, *Mangifera indica* methanolic stem-bark extract. Results are expressed as means ± SEM for three separate experiments at each concentration. For marked significance from controls, a: *p* < 0.05, b: *p* < 0.01, c: *p* < 0.001.

**Figure 3 medicines-06-00015-f003:**
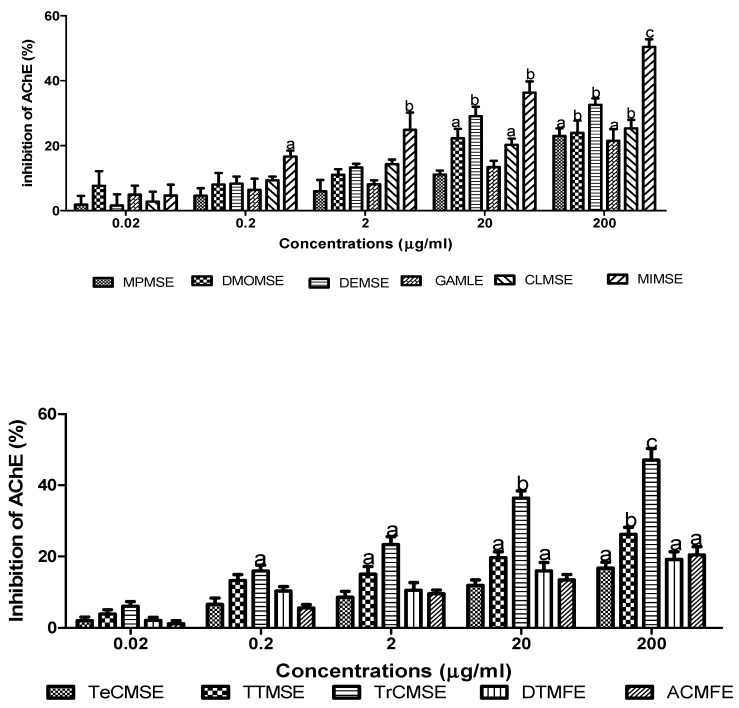
AChE inhibitory activity of plant extracts. Plant inhibition of AChE was measured using a modified Ellman assay, with percentage inhibition of AChE calculated relative to eserine. Results are expressed as means ± SEM for three separate experiments at each concentration. MPMSE, *Musa parasidisiaca* methanolic stem-bark extract; DMOMSE, Defatted *Moringa oleifera* methanolic seed extract; DEMSE, *Dacroydes edulis* methanolic seed extract; GAMLE, *Gnetum africanum* methanolic leaf extract; CLMSE, *Cola lepidota* methanolic seed extract; MIMSE, *Mangifera indica* methanolic stem-bark extract; TeCMSE, *Terminalia catappa* methanolic seed extract; TTMSE, *Tetrapleura tetraptera* methanolic seed extract; TrCMSE, *Tricosanthes cucumerina* methanolic seed extract; DTMFE, *Dennettia tripetala* methanolic fruit extract; ACMFE, *Artocarpus communis* methanolic fruit extract. Results are expressed as means ± SEM for three separate experiments at each concentration. For marked significance from controls, a: *p* < 0.05, b: *p* < 0.01, c: *p* < 0.001.

**Table 1 medicines-06-00015-t001:** Percentage yield, DPPH radical scavenging activity, AChE inhibitory potency, and total phenolic and flavonoid content for the evaluated food sources.

Extracts of Plants	Yield (%)	IC_50_ Concentrations (µg/mL)		Total Phenolic Content (mg GAE/g)	Total Flavonoid Content (mg QUER E/g)
		DPPH Radical Scavenging	AChE Inhibition		
Edible Food					
TeCMSE	7.1	302	834.5	28.45 ± 1.40	21.43 ± 0.98
TrCMSE	59.8	854	201.2	132.65 ± 0.85	64.34 ± 1.43
TTMSE	66.7	205	967.9	25.36 ± 0.87	17.35 ± 1.53
DTMFE	39.4	134	654.3	75.64 ± 1.87	41.24 ± 1.56
ACMFE	3.4	890	576.4	102.45 ± 1.43	69.54 ± 1.73
DEMSE	5.0	138	529.9	95.73 ± 3.62	53.35 ± 2.37
GAMLE	11.2	825	321.9	123.26 ± 2.73	73.26 ± 1.78
CLMSE	13.5	526	438.4	119.63 ± 3.24	68.35 ± 2.65
DMOMSE	21.3	145	657.1	65.15 ± 1.35	31.43 ± 0.83
Non-Edible					
MIMSE	2.1	321	111.9	156.2 ± 2.43	87.35 ± 1.57
MPMSE	7.8	106	619.8	85.36 ± 0.95	42.83 ± 1.24

TeCMSE, *Terminalia catappa* methanolic seed extract; TrCMSE, *Tricosanthes*
*cucumerina* methanolic seed extract; TTMSE, *Tetrapleura tetraptera* methanolic seed extract; DTMFE, *Dennettia tripetala* methanolic fruit extract; ACMFE, *Artocarpus communis* methanolic fruit extract; DEMSE, *Dacroydes edulis* methanolic seed extract; GAMLE, *Gnetum africanum* methanolic leaf extract; CLMSE, *Cola lepidota* methanolic seed extract; DMOMSE, Defatted *Moringa oleifera* methanolic seed extract; MIMSE, *Mangifera indica* methanolic stembark extract; MPMSE, *Musa Parasidisiaca* methanolic stem-bark extract. GAE: gallic acid equivalents; QUER E: quercetin equivalents. Extract was evaluated at least in triplicate across concentration range, and an approximate IC_50_ calculated.

**Table 2 medicines-06-00015-t002:** Correlation variables for AChE IC_50_, DPPH radical scavenging IC_50_ and total phenolic and flavonoid content of the evaluated food sources.

Assessment	AChE Inhibition (IC_50_)	DPPH Radical Scavenging (IC_50_)
AChE inhibition (IC_50_)		R = 0.243 *p* = 0.42
DPPH Radical scavenging (IC_50_)	R = 0.243 *p* = 0.42	
Total phenolics	R = −0.972 *p* = 0.0001	R = 0.488 *p* = 0.127
Total flavonoids	R = −0.84 *p* = 0.0012	R = 0.392 *p* = 0.232

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
