# Peer review of "An Investigation of Potential Sources of Nutraceuticals from the Niger Delta Areas, Nigeria for Attenuating Oxidative Stress"

_medicines, 2019, doi:10.3390/medicines6010015_

Reviewer 1 Report

Your contribution is really important to give evidence about traditional uses of native food plants.

I find it interesting your work being important in the development of further research.

You will find some comments and questions about your work in the file attached.

Author Response

Reviewers’ comments

Reviewer 1: Comments and Suggestions for Authors

Your contribution is really important to give evidence about traditional uses of native food plants.

I find it interesting your work being important in the development of further research.

You will find some comments and questions about your work in the file attached.

We appreciate the kind comments of Reviewer 1 about our research work.  The comments and questions raised in the attached file have now been addressed.

Reviewer 2 Report

The main questions for improving the article which is good and publishable are:

- it would need some  more information on the technique of drying the samples, a fundamental factor for the quality of the tested products;

- the measurement  units of the DPPH assays and reducing power are in microg / mL. On dry  weight of plant, extract, or other? Ask, because in the M & M it is  not specified.

- the analysis of  DPPH and reducing power should be improved, especially as regards the  expression of data, where I only see a comparison with a reference  antioxidant, however tested with very different answers from those of  the extracts, much higher. EC50, by weight, on what  is referred to, dry weight of plant, extract, or other ?;

- as regards to the  analysis of the inhibition of acetylcholinesterase activity, I only see  the data with the inhibition percentage, without a reference, in the M  & M the Authors claim to have used the eserina as a positive  reference, but in the paper I see no data on this, except  for a hint, without real data, in the caption of Figure 3.

Author Response

Reviewer 2: Comments and Suggestions for Authors

The main questions for improving the article which is good and publishable are:

- it would need some  more information on the technique of drying the samples, a fundamental factor for the quality of the tested products;

Response: We appreciate that this information should have been included:  All of the samples were shade or air dried, and this has now been mentioned in the revised manuscript.

- the measurement  units of the DPPH assays and reducing power are in microg/mL. On dry weight of plant, extract, or other? Ask, because in the M & M it is not specified.

Response: the measurement units were IC50 concentrations (µgram/mL). This was not specified in the materials and methods because we made reference to an earlier more detailed publication (reference 56).

- the analysis of  DPPH and reducing power should be improved, especially as regards the  expression of data, where I only see a comparison with a reference  antioxidant, however tested with very different answers from those of  the extracts, much higher. EC50, by weight, on what  is referred to, dry weight of plant, extract, or other ?

Response: DPPH and reducing power were conducted via a direct comparison to usually employed standards, that also acted as positive controls. This direct comparison to Vitamin E and Vitamin C is commonplace throughout these type of analyses (for example, refer to Reference 56 and references therein).  Since we are not measuring the efficacy of a drug, determination of the effective concentration dose (EC50) has no value.  Rather, we are quantifying the ability of the plant agents to mitigate free radicals and limit oxidation; hence the ability to act as an inhibitor (expressed as an IC50 concentration) was determined.

-as regards to the analysis of the inhibition of acetylcholinesterase activity, I only see  the data with the inhibition percentage, without a reference, in the M  & M the Authors claim to have used the eserina as a positive  reference, but in the paper I see no data on this, except  for a hint, without real data, in the caption of Figure 3.

Response: The use of eserine (physostigmine) as a reversible and relatively potent inhibitor of AChE is commonplace, indeed, physostigmine was one of the first drugs developed as an anti-cholinesterase treatment for Alzheimer’s disease.  We describe the use of eserine in the materials and methods section (“The AChE used was either electric eel at 1 mg/mL (Sigma, Poole, UK) or that present within rat brain homogenate (prepared at 10% (w/v) according to the procedure of Carter et al.[58,59] that had been diluted 1:10 in 10 mM Tris-HCl pH 8.0 for assays”, and also reference our previous manuscript (reference 56).  Even at the lowest concentration of eserine used, 0.02 µg/ml (≈72 nM), AChE is essentially inhibited by 100%, since the IC50 concentration of eserine is ≈14 nM (again refer to Reference 56).  Hence for all other data points in Figure 3 (0.2, 2, 20, and 200 µg/mL), the use of an increased eserine concentration (above 0.02 µg/mL) did not induce further AChE inhibition.  We appreciate that this could be clearer, and therefore in the materials and methods section we have added the following text: Eserine was employed at 0.02 µg/ml as a positive control.  At this concentration (or above), eserine inhibits AChE to ≈100% [56, results not included].

Reviewer 3 Report

In this research paper, the authors investigated seed and fruit extracts of various plants from dietary sources in Niger Delta area of Nigeria. They mainly focused on the evaluation of antioxidant activity of those extracts through determination of DPPH radical scavenging and reducing power capacity, their inhibition of Acetylcholinesterase (AChE) from electric eel or rat brain homogenate, and determination of total phenolic and flavonoid contents of those extracts. In conclusion, they found correlation between AChE inhibition and total phenolic and flavonoid contents of those extracts. The reviewer considers that this manuscript provide a topic of interest to the audiences in this field with up-to-date information. Once the following minor issues are addressed, it can be moved forward for consideration of publication in Medicines.

1. In Figure 1, it would be more logic to compare the DPPH scavenging activity of the tested extracts to Vitamin E at each concentration. In another word, the DPPH scavenging activity for Vitamin E should be 100% for each concentration tested. Similarly, in Figure 2, it would be more logic to compare the reducing capacity of the tested extracts to Vitamin C at each concentration.

2. In Figure 3, why the data for positive control Eserine were not presented? It should be consistent with previous figures since Vitamin E and C are presented in Figure 1 and 2, respectively.

3. In the method section, line 184, what is the rationale for the selection of 1.56, 0.78,…and 0.0975 mg/ml as concentrations for Vitamin E? It should be explained since the dilution starts from 1.56, an odd number.

4. In the method section, what is the concentration of Vitamin C used as positive control to evaluate reducing power capacity of those extracts?

5. Format issue. Make sure the titles for X and Y axis are presented in the same font for the graphs in Figure 2. Same suggestion goes to Figure 3 as well.

6. Minor typo and grammar issues need to be corrected and checked one more time.  For example, Line 204 – 205, “the AChE used was either electric eel…” can be replaced by “the AChE used was either from electric eel…” to avoid confusion. Line 369, “Out study…” should be “Our study…”. Figure 1, for the title of Y axis on the second graph, it should be “DPPH scavenging activity (%)” not “DPPH scavengeing activity (%)”.

Author Response

Reviewer 3: Comments and Suggestions for Authors

In this research paper, the authors investigated seed and fruit extracts of various plants from dietary sources in Niger Delta area of Nigeria. They mainly focused on the evaluation of antioxidant activity of those extracts through determination of DPPH radical scavenging and reducing power capacity, their inhibition of Acetylcholinesterase (AChE) from electric eel or rat brain homogenate, and determination of total phenolic and flavonoid contents of those extracts. In conclusion, they found correlation between AChE inhibition and total phenolic and flavonoid contents of those extracts. The reviewer considers that this manuscript provide a topic of interest to the audiences in this field with up-to-date information. Once the following minor issues are addressed, it can be moved forward for consideration of publication in Medicines.

1. In Figure 1, it would be more logic to compare the DPPH scavenging activity of the tested extracts to Vitamin E at each concentration. In another word, the DPPH scavenging activity for Vitamin E should be 100% for each concentration tested. Similarly, in Figure 2, it would be more logic to compare the reducing capacity of the tested extracts to Vitamin C at each concentration.

Response: We can appreciate the suggestion of this reviewer of setting the vitamin E at each extract assay concentration as 100% (Figure 1) or vitamin C (Figure 2) at 100%, and then making a direct quantitation for the extracts against these values.  However, there are caveats with this type of data display.  Firstly, by also increasing the vitamin E and vitamin C concentrations, a dose effect relationship for each of the vitamins is evident: increasing vitamin E produces more radical scavenging and increasing vitamin C produces more reducing capacity.  This additional information and reinforcement of the use of these agents as both standards and positive controls would otherwise be lost.  Secondly, the representation of data would require a new graph for each concentration data point, resulting in the generation of 13 Figures for Figure 1 and Figure 2 data.  We believe this would not be beneficial to the reader, as by displaying the data over 1 graph for Figure 1 and two graphs for Figure 2, it is more easily digested by a reader and the dose-response nature of the agents is more easily followed.

 2. In Figure 3, why the data for positive control Eserine were not presented? It should be consistent with previous figures since Vitamin E and C are presented in Figure 1 and 2, respectively.

Response: We can appreciate this comment as it overlaps with that mentioned by Reviewer #2.  We would refer Reviewer #3 to our response to Reviewer #2 that even at the lowest eserine concentration employed (0.02 µg/ml) there was ≈100% inhibition of AChE.  Hence, we considered it to be superfluous to include 100% inhibition histograms for eserine at all of the data points; it would make the graphs look more crowded and would not have provided any additional information.

 3. In the method section, line 184, what is the rationale for the selection of 1.56, 0.78,…and 0.0975 mg/ml as concentrations for Vitamin E? It should be explained since the dilution starts from 1.56, an odd number.

Response: We appreciate that this does seem an odd starting concentration of 1.56 mg/mL from which to begin the standard dilutions for Vitamin E, rather than a uniform number such as 2 mg/mL.  This simply reflects a concentration range that provides a usable OD change when assessed spectrophotometrically.  Additionally, the concentration range was in keeping with previous publications (Reference 56, and our other subsequent publications: Nwidu et al (2018a) Medicines 5(3), pii; E71; Nwidu et al (2018b) Malaysian Journal of Medical Sciences 25, 27-39), enabling a reader to make direct comparisons to plant extracts examined in other publications.

 4. In the method section, what is the concentration of Vitamin C used as positive control to evaluate reducing power capacity of those extracts?

Response: We appreciate that this should have been included and was also flagged by Reviewer #1. Details of the use of ascorbic acid as the reference substrate and the concentration range employed (0.3, 0.6, 0.9, 1.2, 1.5, and 3 mg/mL) have now been included into the materials and methods section.

 5. Format issue. Make sure the titles for X and Y axis are presented in the same font for the graphs in Figure 2. Same suggestion goes to Figure 3 as well.

Response: This is an oversight by us, and all Figures have been revised to ensure that the text used is consistent throughout.

6. Minor typo and grammar issues need to be corrected and checked one more time.  For example, Line 204 – 205, “the AChE used was either electric eel…” can be replaced by “the AChE used was either from electric eel…” to avoid confusion.

Response: We have made the change suggested above and checked through the manuscript for other typo and grammatical issues.

Line 369, “Out study…” should be “Our study…”. Figure 1, for the title of Y axis on the second graph, it should be “DPPH scavenging activity (%)” not “DPPH scavenging activity (%)”.

Response: This typographical error was amended, and likewise errors in the Figures have been corrected.
